# Kanamycin and Ofloxacin Activate the Intrinsic Resistance to Multiple Antibiotics in *Mycobacterium smegmatis*

**DOI:** 10.3390/biology12040506

**Published:** 2023-03-27

**Authors:** Aleksey A. Vatlin, Olga B. Bekker, Kirill V. Shur, Rustem A. Ilyasov, Petr A. Shatrov, Dmitry A. Maslov, Valery N. Danilenko

**Affiliations:** 1Institute of Ecology, Peoples’ Friendship University of Russia (RUDN University), 117198 Moscow, Russia; vatlin_alexey123@mail.ru; 2Laboratory of Bacterial Genetics, Vavilov Institute of General Genetics Russian Academy of Sciences, 119333 Moscow, Russia; obbekker@mail.ru (O.B.B.); shurkirill@gmail.com (K.V.S.); valerid@vigg.ru (V.N.D.); 3Laboratory of Molecular Genetics, Bashkir State Agrarian University (BSAU), 450001 Ufa, Russia; 4Phystech School of Biological and Medical Physics, Moscow Institute of Physics and Technology (National Research University), 141701 Dolgoprudny, Russia; shatrov.pa@phystech.edu; 5Division of Gastroenterology and Hepatology, Department of Medicine, Stanford University School of Medicine, Stanford, CA 94305, USA; dmaslov@stanford.edu

**Keywords:** *Mycobacterium smegmatis*, cross-resistance, antibiotics

## Abstract

**Simple Summary:**

Tuberculosis is one of the deadliest bacterial diseases in the world. Drug resistant strains of *Mycobacterium tuberculosis* pose a particular threat to global healthcare. In this work, we analysed the effect of micro-concentrations of antibiotics, which can be found in the environment, on the occurrence of induced resistance in mycobacteria. It was shown that concentrations of antibiotics (kanamycin, ofloxacin) that do not affect bacterial growth can induce the expression of resistome genes and lead to increased levels of resistance to a range of antibiotics.

**Abstract:**

Drug resistance (DR) in *Mycobacterium tuberculosis* is the main problem in fighting tuberculosis (TB). This pathogenic bacterium has several types of DR implementation: acquired and intrinsic DR. Recent studies have shown that exposure to various antibiotics activates multiple genes, including genes responsible for intrinsic DR. To date, there is evidence of the acquisition of resistance at concentrations well below the standard MICs. In this study, we aimed to investigate the mechanism of intrinsic drug cross-resistance induction by subinhibitory concentrations of antibiotics. We showed that pretreatment of *M. smegmatis* with low doses of antibiotics (kanamycin and ofloxacin) induced drug resistance. This effect may be caused by a change in the expression of transcriptional regulators of the mycobacterial resistome, in particular the main transcriptional regulator *whiB7*.

## 1. Introduction

Tuberculosis remains the leading cause of death from an infectious disease among adults worldwide, with more than 10 million people becoming newly sick from tuberculosis each year. Drug resistance (DR) in *Mycobacterium tuberculosis* is one of the main problems in anti-tuberculosis (anti-TB) therapy. Drug-resistant forms of tuberculosis are currently on course to be the world ‘s deadliest pathogens, responsible for a quarter of deaths due to antimicrobial resistance. It has traditionally been thought that the selection of resistant microorganism strains occurs mainly at high therapeutic levels of antibiotics [1,2]. Recently, there have been more and more reports in the literature about the presence of subinhibitory concentrations of antibiotics, which are orders of magnitude lower than the standard known minimal inhibitory concentrations (MICs) [3,4,5,6]. Subinhibitory concentrations are capable of accelerating the emergence of drug-resistance in bacterial strains [7,8,9], and the selection occurs at significantly lower concentrations than the MICs. However, the selection is possible at lower concentrations, which is known as the susceptible minimum persistent concentration (MIPC, sub-MSC selective window) [10]. The selection of resistant strains at MIPCs occurs due to differences in growth rates between cells with different levels of antibiotic tolerance. Studies have shown that resistant bacteria can be selected at concentrations several hundred times lower than lethal concentrations [11]. It is important to note that MIPCs can be equal to the concentrations of antibiotics found in natural environments, e.g., ciprofloxacin induced resistance at concentrations found in the aquatic environment [10,11].

To date, the processes of induction of drug resistance by subinhibitory concentrations in pathogenic microorganisms have been partially described [3,4,5,6]. 

Actinobacteria are known to possess a large reservoir of genes involved in intrinsic drug resistance—the resistome [12]. Mycobacteria are not an exception. The *whiB7* is a key regulator of the resistome, which can provide a fairly high level of resistance to antimicrobial agents and is not associated with the appearance of mutations in target genes [13,14]. This system provides a cellular response to the exposure of antibiotics by different mechanisms, such as their efflux from the cell, antibiotic molecule inactivation, and target modification [15]. Exposure to low doses of antibiotics may not only lead to an increase in the level of resistance to the same class of antibiotics [16] but may also increase the frequency of the selection of strains resistant to other classes of antibiotics [17].

The US Food and Drug administration gives information that was current as of 17 January 2023: «Tolerances for residues of new animal drugs in food», which includes in particular the groups of fluoroquinolones, aminoglycosides, tetracyclines et al. [18].

Antibiotic pollution is one of the key routes by which bacteria are able develop resistance to life-saving medicines, rendering them ineffective for human use. In 2019, the largest global study on the subject found two-thirds of hundreds of test sites in rivers around the world, from the Thames to the Tigris in 72 countries, were awash with dangerously high levels of antibiotics [19].

We used *Mycobacterium (Mycolicibacterium) smegmatis* mc2 155^−2^ for *Mycobacterium tuberculosis* as an adequate and commonly used model organism to analyse antibiotic resistance. In a study by Altaf et al., *M. smegmatis* was resistant to approximately 50% of anti-mycobacterial compounds listed in the Library of Pharmacologically Active Compounds (LOPAC) that were detected as active against *M. tuberculosis* [20]. The “Rule of five” by Lipinski formed the basis of the library of low molecular weight compounds Lopak. The inability of many small organic molecules to dissolve in aqueous media and/or engage biological targets in a specific, stoichiometric manner was among the primary developmental failures associated with early combinatorial libraries. Outcomes from screening these libraries were plagued with false positives (often attributed to aggregation events) or overly lipophilic leads with limited optimization potential. Lipinski and co-workers were among the first to perform an extensive cheminformatics assessment of solubility and permeability. This analysis showed that the 90th percentile of a set of >2000 orally bioavailable agents with acceptable solubility and permeability had a MW under 500, fewer than 5 H-bond donors and 10 H-bond acceptors, and a cLogP value less than 5 [21]. The 50% antibiotic resistance differences between *M. smegmatis* and *M. tuberculosis* can be explained by the presence of a wider set of instruments in *M. smegmatis* that can neutralize various drugs, though it has a similar cell wall, over 2000 homologous genes, and shows similar metabolic cellular processes [22]. 

In this study, we attempted to investigate the most promising mycobacterial resistome genes involved in drug resistance and in a response to antibiotic-induced stress after induction by maximum non-inhibiting concentrations of antibiotics, which are significantly lower than standard MICs. 

## 2. Materials and Methods

### 2.1. Bacterial Strains, Media

*Mycobacterium smegmatis* mc2 155^−2^ was cultured in Lemco-TW broth (5 g/L Bacto-peptone, Becton–Dickinson and Company, USA; 5 g/L Lab-Lemco powder, Oxoid, Basingstoke, UK; 5 g/L NaCl; 0.05% *v*/*v* Tween-80) at 200 rpm and 37 °C, while Soyabean casein digest agar M290 (HiMedia, India) was used as the solid medium.

### 2.2. MIC Determination

The MICs of the studied antibiotics on *M. smegmatis* were determined in liquid medium. A standard procedure involving serial two-fold dilutions of the tested antibiotics was used to determine the MICs in liquid Lemco-Tw medium. The concentration fully inhibiting bacterial growth as compared to the antibiotics-free control sample was defined as the MIC. *Mycobacterium smegmatis* mc2 155^−2^ was cultured in Lemco-TW overnight, then diluted in the proportion of 1:200 in fresh medium (to approximately OD600 = 0.05). A volume of 196 μL of the diluted culture was poured into sterile nontreated 96-well flat-bottom culture plates (Eppendorf, Germany) and 4 μL of serial two-fold dilutions of the tested antibiotics were added to the wells. The plates were incubated at 37 °C and 200 rpm for 48 h. The MIC was determined as the lowest concentration of the compound with no visible bacterial growth and OD600 measured on the DTX 880 Multimode Detector (Beckman Coulter, Brea, CA, USA) did not differ from the OD600 of the medium without *M. smegmatis*. The last three independent repetitions of the experiment were performed in increments of 0.1 µg/mL for kanamycin and streptomycin and in increments of 0.01 µg/mL for ofloxacin and tetracycline. The aliquots of *M. smegmatis* mc2 155^−2^ culture from the 96-well flat-bottom culture plates was plated on the soybean casein digest agar (M290, Himedia, India) to confirm the purity of the culture *M. smegmatis* mc2 155^−2^. Afterwards, the MIC values, presented in the liquid medium, were checked in the agar medium. The value of MIC was ascertained as the lowest concentration that inhibited the growth of 99% of colony forming units (CFU). 

### 2.3. Search for Maximum Non-Inhibiting Antibiotics Concentrations

The overnight Lemco-Tw culture of *M. smegmatis* (OD_600_~1.8–2) was inoculated in fresh Lemco-Tw broth (1:200 *v*/*v* dilution) supplemented with serial two-fold dilutions of tested antibiotics (kanamycin, streptomycin, ofloxacin, and tetracycline). The 96-well flat-bottom plates (Cellstar, Greiner bio-one, Kremsmünster, Austria) with 200 μl of the culture medium were used in this assay. After 24 h of incubation at 200 rpm and 37 °C, the OD_600_ was measured on a DTX 880 Multimode Detector (Beckman Coulter, Brea, CA, USA). The maximum non-inhibiting antibiotics concentrations was considered as the maximum concentration at which CFU *M. smegmatis* grown with an antibiotic was approximately the same CFU *M. smegmatis* grown without antibiotics.

### 2.4. Drug Susceptibility Testing

The overnight culture of *M. smegmatis* was diluted in fresh Lemco-Tw medium, supplemented with an antibiotic in the maximum non-inhibiting concentration, with an initial OD_600_ = 0.05 in 10 mL. The cultures were incubated at 200 rpm and 37 °C overnight until OD_600_ = 1.2. Afterwards, drug susceptibility was assessed by paper-disc assay: the culture of *M. smegmatis* was diluted 1:9:10 (culture:water:M290 medium) and seeded over the base agar layer on Petri dishes. The culture medium was supplemented with the same amount of antibiotic used for induction in liquid medium. The HiMedia Laboratories Pvt. Ltd. (Maharashtra, India)—discs (Kirby–Bauer Disk Diffusion Susceptibility Test Protocol) were used for most antibiotics (netilmicin 10 μg/disc, meropenem 10 μg/disc, imipenem 10 μg/disc, spiramycin 100 μg/disc, norfloxacin 10 μg/disc, ofloxacin 5 μg/disc, linezolid 10 μg/disc, kanamycin 30 μg/disc, oxytetracycline 30 μg/disc, azithromycin 15 μg/disc, levofloxacin 5 μg/disc, ciprofloxacin 1 μg/disc, tetracycline 30 μg/disc, lomefloxacin 10 μg/disc), while the rifampicin 100 μg/disc was manually impregnated on sterile paper discs. The plates were incubated for 2 days at 37 °C until the bacterial lawn was fully-grown. Growth inhibition halos were measured to the nearest 0.1 mm (halo area around the disk was measured with a digital calliper and analysed in a Mega Bio-Print 3020-WL/LC/20M X-Press, Vilber Lourmat Gel Documentation System). The experiments were carried out as five repeats; the average diameter and standard deviation (SD) were calculated. A significant difference in the growth inhibition halo was considered to be the ones that had no intersection with the SDs of the control. As a control for the absence of a synergic effect of antibiotics, we determined the halo diameter around the standard antibiotic discs soaked with the antibiotics used for induction. The antibiotics used for induction were added to the discs in the induction concentration. Halo diameter was determined on *M. smegmatis* grown without antibiotics [23]. The criterion for selecting positive results in Section 3.3 was the absence of intersections of the standard deviations of the diameters of the zones of growth inhibition for samples of *M. smegmatis* grown on a medium with an antibiotic (inductor) and on a medium without it. We used CLSI standards that contain information about disk diffusion (M02) and dilution (M07, M11) test procedures for bacteria [23] and Laboratory Practices in Microbiology [24].

### 2.5. Mycobacterial RNA Isolation and Real-Time qPCR

Cells from the 10 mL culture were harvested by centrifugation for 10 min at 3000× *g* and 4 °C, washed twice with 3 mL of RNAprotect Bacteria Reagent (Qiagen, Germantown, MD, United States) for RNA stabilization. The cells of M. smegmatis were homogenized in ExtractRNA reagent (Evrogen, Moskow, Russia), followed by phenol (pH = 4.5)-chloroform-isoamyl alcohol (25:24:1) purification and precipitation with isopropanol (2:1, *v*/*v*). The remaining genomic DNA was removed by DNAse I, Amplification grade (Invitrogen, USA). A volume of 50 ng of total RNA was used for cDNA synthesis by iScript Select cDNA Synthesis Kit (Bio-Rad, Berkeley, CA, USA). A sample of 1 ng of cDNA was used for real-time qPCR with the qPCRmix-HS SYBR kit (Evrogen, Russia) on a CFX96 Touch machine (Bio-Rad, USA). The CFX Manager V 3.1 (Bio-Rad, USA) was used to analyse the qPCR results: relative normalized expression of three biological replicates was calculated as ΔΔCq [25] and the genes *sigA* and *polA* were used as a reference. The primers were picked by primer-BLAST [26] for qPCR (Appendix A, Table A1).

## 3. Results

### 3.1. Minimal Inhibiting Antibiotic Concentrations Determination for M. smegmatis

Determination of antibiotic concentrations is necessary to further determine non-inhibitory concentrations of antibiotics to study the induction of resistance to other antibiotics classes used in therapy. The following antibiotics were used to determine MIC: kanamycin and streptomycin (aminoglycosides), tetracycline (tetracyclines), and ofloxacin (fluoroquinolones). We measured the MICs of the antibiotics in liquid media on *M. smegmatis* −2. The results are presented in Table 1. The MIC of the antibiotics on *M. smegmatis* did not differ much from the MIC of the *M. tuberculosis* strain, which suggests the presence of similar mechanisms of action on the cell.

### 3.2. The Determination of Maximum Non-Inhibiting Concentrations

For the planned study, we needed to determine the maximum subinhibitory antibiotic concentrations that would not affect the growth dynamics of *M. smegmatis* in order to avoid any possible synergistic effect such as the occurrence of drug cross-resistance. We selected the following antibiotics as possible inducers: streptomycin, kanamycin, ofloxacin, and tetracycline. These antibiotics, which belong to different chemical classes, are used for antimicrobial therapy, including anti-TB therapy, and are considered basic for the pertaining chemical classes. The effects of different dilutions of antibiotics on the growth speed of *M. smegmatis* are shown in Table 2.

We selected the following concentrations of antibiotics for further experiments as the maximum non-inhibiting: kanamycin—0.03 µg/mL, streptomycin—0.016 µg/mL, ofloxacin—0.08 µg/mL, tetracycline—0.015 µg/mL. 

### 3.3. Evaluation of the Intrinsic Drug Cross-Resistance Induction

We evaluated the drug sensitivity of cultures of *M. smegmatis* pretreated with the maximum non-inhibitory concentrations of various antibiotics such as kanamycin, streptomycin, ofloxacin and tetracycline by the paper-disc assay to investigate how the exposure to low doses of antibiotics induces intrinsic resistance. The culture grown in a medium without any antibiotics was used as a control. Figure 1 shows the results of significant changes in the drug susceptibility of the culture of *M. smegmatis* after exposure to different antibiotics (Table A2). 

The obtained results allowed us to divide the inducers into three groups: (1) high-specific inducers (streptomycin) that lead to a specific response triggering resistance to only rifampicin and probably involving a limited activation of antibiotic resistance factors; (2) inducers with moderate specificity (kanamycin and ofloxacin); (3) inducer with low specificity (tetracycline). We have shown that treatment with maximum non-inhibiting concentrations, which do not affect cell growth, can lead to a change in the level of cell resistance to a variety of antibiotics.

The pre-treatment of the culture of *M. smegmatis* cells with tetracycline led to an increase in the drug resistance to virtually all the classes of antibiotics: fluoroquinolones, macrolides, aminoglycosides, linezolids, and tetracyclines (Figure 1). There is an indirect explanation of these results: tetracycline specifically induces the tetracycline inducible TetR-family regulators (TFRs) and thus initiates the transcription of a plethora of different genes mediating drug resistance. The genome of *M. tuberculosis* harbours at least 49 genes encoding TFRs. After entering the cell, tetracycline initiates the transcription of many different genes, including those that mediate drug resistance [31].

### 3.4. Selection of Candidate Genes Potentially Involved in Cross-Resistance

As the intrinsic cross-resistance mainly relies upon the action of various transcription factors [13], we conducted a search for genes that are potentially involved in the implementation of cross-resistance in *M. smegmatis*, based on the published data and the observed resistance phenotype. The known transcriptional regulators as *whiB7*, *tetR*, *araC*, *lfrR*, *ltmA*, *marR*, *mtrA*, *napM*, *rbpA* of various chemical classes, whose participation in the formation of resistance to antibiotics has been shown, were chosen for the study as we aimed to identify the transcriptional regulators, triggering the induction of resistance in the cell at maximum non-inhibiting concentrations. Their regulated genes underlying many events, such as antibiotics production, osmotic stress, efflux pumps, multidrug resistance, metabolic modulation, and pathogenesis [32,33]. Thus, we selected global multidrug regulators with partially unknown functions that participate in the “transcriptional drug response” (Table 3).

We aimed to reveal any possible correlation between the expression level of these genes under induction by various antibiotics and the observed intrinsic cross-resistance [30].

The unspecific action of tetracycline that activates many TetR transcription factors [31,43] forced us to exclude it from further analysis. 

### 3.5. Study of Gene Expression of Intrinsic Cross-Resistance of M. smegmatis

We conducted transcriptional analysis of the selected genes (Table 3) to assess their involvement in induced drug cross-resistance. We used the same conditions as in Section 3.3 to study the changes of transcription level of the selected genes. Streptomycin, kanamycin and ofloxacin were added to the culture of *M. smegmatis* in 0.016 µg/mL, 0.03 µg/mL, and 0.08 µg/mL, respectively. After 24 h of the growth (log-phase) of *M. smegmatis*, RNA was extracted. The relative expression levels of the studied genes are shown in Figure 2.

Genes were considered to be significantly differentially expressed if their expression changed > 2-fold with a *p*-value < 0.05, compared to their expression without induction. Genes subject to this criterion are listed in Table 4.

We observed a significant increase in *whiB7* transcription by kanamycin and ofloxacin; however, ofloxacin showed itself as an inhibitor of transcription of at least six genes. We did not observe any significant changes in transcription levels caused by streptomycin. 

## 4. Discussion

The emergence of drug resistance in pathogenic microorganisms, particularly in *M. tuberculosis*, is one of the main challenges facing medicine. It is known that in addition to acquired drug resistance (DR), the natural DR system plays a huge role in cell survival and can cause a fairly high level of resistance to antimicrobial agents, which is not associated with the appearance of mutations in target genes [2]. Environment contamination by the antibiotics may play an important role in the formation of drug resistance in bacteria. The widespread use of antibiotics leads to the creation of antibiotic concentration gradients in the environment, exposing bacteria to sub-inhibitory concentrations of drugs, and contributing to the development of antibiotic resistance [44,45,46].

Several terms are used in the literature to describe concentrations less than the MIC. One of them is the minimal antibiotic concentration (MAC), which is the lowest concentration of an antibacterial agent that produces a decrease of 1 log in the number of organisms/mL as compared with a control culture in drug-free medium [47] or subinhibitory antibiotic concentrations (sub-MICs), which leads to one log10 decrease in a bacterial population compared to the control [48]. Additionally, the term minimal selective concentration (MSC) is used, where the resistant strain is enriched over the susceptible [10]. The maximum non-inhibiting antibiotics concentrations used in this work, were determined as those which do not affect a change in the number of colony forming units of the cell culture in comparison with the antibiotic and without it. These concentrations are lower than sub-MICs, MACs, and even MSCs.

In our work, we analysed changes in drug susceptibility and expression levels of known resistome genes in *M. smegmatis* upon exposure to low doses of antibiotics, ones not affecting bacterial growth. It was shown that streptomycin, kanamycin, ofloxacin and tetracycline (in the maximum non-inhibiting concentrations) can cause a cell response leading to the development of cross-DR. However, we excluded the inductor, tetracycline, from further work to avoid a non-specific response. It is known that tetracycline affects the functioning of the TetR-family transcriptional regulators (there are about 154 of them in the genome of *M. smegmatis* according to its annotation [accession number: PRJNA57701]), which control many DR genes in mycobacteria [43].

After analysing the transcriptional profile of genes following the addition of maximum non-inhibiting concentrations of an antibiotic, genes can be divided into two types: an increase and a decrease in their level of expression. Thus, an increase in the gene expression of the *whiB7* transcription factor was observed in the case of the inducers kanamycin and ofloxacin. According to the literature data, WhiB7 is a global regulator of natural drug resistance, which agrees with obtained results. However, we did not find a significant increase in *whib7* gene expression upon streptomycin induction. This shows a potentially specific WhiB7 response to antibiotics [32]. 

In addition to an increase in the expression of a number of genes, we showed a decrease in the expression activity of genes upon induction with ofloxacin (Table 3). Thus, we found that there is a decrease in the expression of the repressor protein genes TetR, MarR, and MtrA [32,49]. An interesting fact was that a decrease in the level of expression of the *araC*, which is a transcriptional activator of cellular transporters and removes antibiotics from the cell, was observed [33]. Particular attention should be paid to a large decrease in the expression of the transcriptional regulator *napM*. We did not find an increase in drug resistance to rifampicin, as expected in the work [41]. It is possible that the resistance to antibiotics described in [41] is formed during long-term incubation of cells in a medium with an antibiotic. It can also be assumed that the repression of other genes (NapM is a repressor of about 121 genes), which products lead to the formation of the corresponding resistance phenotype, stops in this case.

The data obtained correlate with the works that described the mechanisms of drug resistance activation after exposure to maximum non-inhibiting concentrations. In *M. tuberculosis*, it was found that after initial treatment with low doses of antibiotics of the aminoglycoside group and subsequent exposure to aminoglycoside antibiotics on these cells, the level of drug resistance to them increases [16]. Additionally, in mycobacteria, it has been shown that subinhibitory concentrations of ciprofloxacin can increase the frequency of selection of strains resistant to other classes of antibiotics [17].

It is assumed that one of the ways for the emergence of increased resistance in infectious strains is the activity of the natural drug resistance system, which can be induced by antibiotics used in agriculture in the production of meat and dairy or other agricultural products, as well as the uncontrolled use of antibiotics by the population [50,51,52].

## 5. Conclusions

We have shown that the treatment of *M. smegmatis* mc2 155^−2^ with concentrations of antibiotics (kanamycin and ofloxacin) that does not affect bacterial growth causes the induction of drug resistance to several antibiotics. According to the data, the U.S. FDA Acceptable daily intake (ADI) for total tetracycline residues (chlortetracycline, oxytetracycline, and tetracycline) is 25 µg/kg of body weight per day (FDA: Sec.556.720) [18]. According to our data, tetracycline at a concentration of 15 µg/L induces resistance to nine antibiotics used in clinical practice. This effect may be caused by a change in the expression of a number of transcriptional regulators of the mycobacterial resistome. Therefore, the study of the mechanisms arising from the impact of micro-doses of antibiotics on the cell is an important task.

## Figures and Tables

**Figure 1 biology-12-00506-f001:**
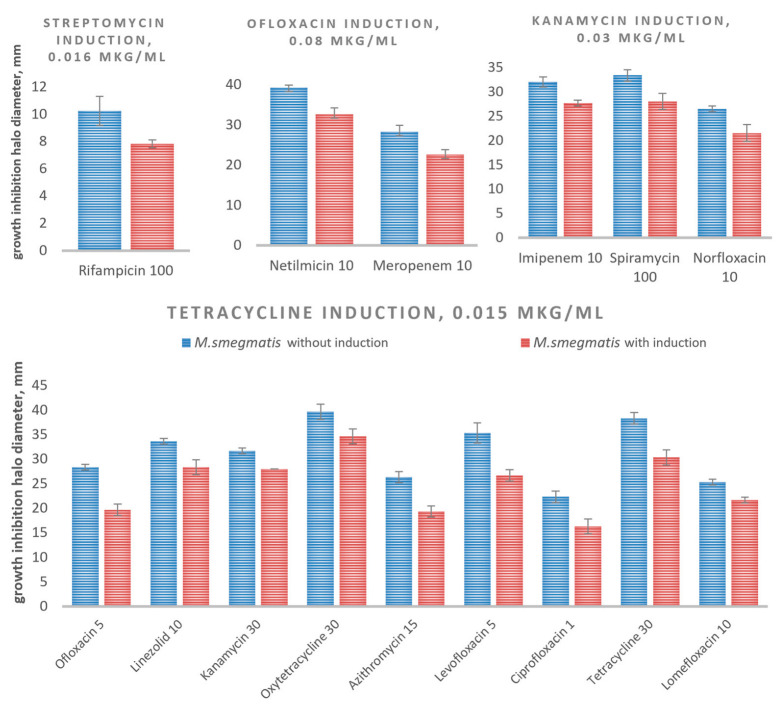
Diameters of growth inhibition halos produced by different antibiotics with and without induction by low doses of antibiotics on cultures of *M. smegmatis*. The columns represent the mean value, while the error-bars represent standard deviations (SD) from five independent experiments.

**Figure 2 biology-12-00506-f002:**
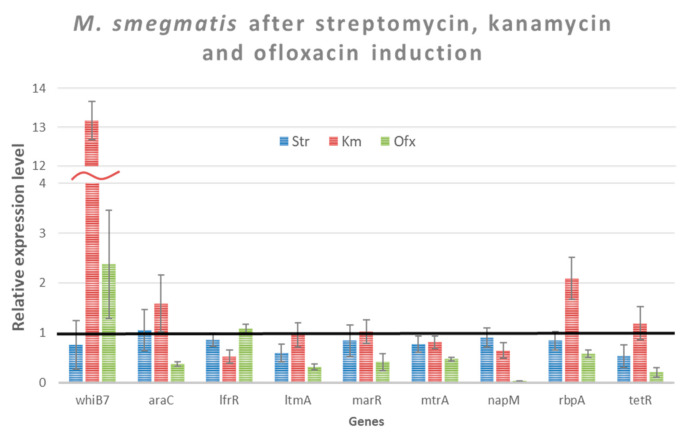
The relative level of the studied genes expression in cells of *M. smegmatis* cultivated in the presence of antibiotics. Expression of the studied genes in the absence of antibiotics was considered equal to 1 (black line—*M. smegmatis* control without induction). The error bars represent the standard deviation calculated from three independent replicates.

**Table 1 biology-12-00506-t001:** The MICs of the antibiotics on *M. smegmatis* and *M. tuberculosis*. Data for *M. smegmatis* were obtained in this paper. Data for *M. tuberculosis* were taken from open sources.

Antibiotics	MIC, μg/mL		Ref.
	*M. smegmatis* mc2 155^−2^	*M. tuberculosis*	
Kanamycin	3.2 ± 0.2	4	[27]
Streptomycin	0.8 ± 0.1	1	[28]
Ofloxacin	0.32 ± 0.03	1.25	[29]
Tetracycline	0.06 ± 0.01	0.55	[30]

**Table 2 biology-12-00506-t002:** The determination of target concentrations of antibiotics, which do not inhibit growth of *M. smegmatis*. The OD_600_ at 0 h of incubation (start point) was 0.01 for all antibiotics. The experiment was repeated in three independent biological replicates—the table shows the average values of optical density. The concentration at which the optical density in the control and experimental groups was the same after 24 h was considered to be the maximum non-inhibiting concentration, * *M. smegmatis* mc2 155 without antibiotics.

Chemical Class	Antibiotic	Proportion of MIC, Concentration, µg/mL	OD_600_ after 24 h of Incubation
Aminoglycosides	Kanamycin	Control Sample *	**~0.2**
		MIC—3.2	~0.012
		½ MIC—1.6	~0.06
		1/60 MIC—0.05	~0.17
		**1/120 MIC—0.03**	**~0.21**
	Streptomycin	Control Sample *	**~0.21**
		MIC—0.8	~0.01
		½ MIC—0.4	~0.023
		1/25 MIC—0.2	~0.07
		**1/50 MIC—0.016**	**~0.2**
Fluoroquinolones	Ofloxacin	Control Sample *	**~0.25**
		MIC—0.32	~0.02
		**1/4 MIC—0.08**	**~0.25**
Tetracyclines	Tetracycline	Control Sample *	**~0.22**
		MIC—0.06	~0.02
		½ MIC—0.03	~0.12
		**1/4 MIC—0.015**	**~0.23**

**Table 3 biology-12-00506-t003:** Genes from *M. smegmatis* selected for study as they are involved in intrinsic drug cross-resistance.

Gene Name (Locus Tag)	Predicted Function	Drug Resistance Phenotype	Ref.
*whiB7* (MSMEG_1953)	Transcriptional factor	Aminoglycosides, macrolides, tetracyclines, fluoroquinolones, phenicols, β-lactams	[34,35]
*tetR* (MSMEG_4022)	Transcriptional factor	Rifampin	[32,36]
*araC* (MSMEG_0307)	Transcriptional factor	Rifampin, kanamycin, chloramphenicol	[33]
*lfrR* (MSMEG_6223)	Transcriptional factor	Fluoroquinolones	[37]
*ltmA* (MSMEG_6479)	c-di-GMP-depended Transcriptional factor	Rifampin, isoniazid	[38]
*marR* (MSMEG_6508)	Transcriptional factor	Isoniazid, Rifampin, ethambutol, kanamycin	[39]
*mtrA* (MSMEG_1874)	Transcriptional factor	Isoniazid, streptomycin, Rifampin	[40]
*napM* (MSMEG_6903)	Transcriptional factor	Rifampin, ethambutol	[41]
*rbpA* (MSMEG_3858)	Transcriptional factor	Rifampin	[42]

**Table 4 biology-12-00506-t004:** A 2-fold or a higher change in the expression level of genes under the antibiotic stress.

Antibiotic	Transcription Level Increase (Fold)	Transcription Level Decrease (Fold)
Kanamycin	*whiB7* (13.2)	
	*rbpA* (2.08)	
Ofloxacin	*whiB7* (2.4)	*napM* (27.42)
		*tetR* (4.88)
		*araC* (2.67)
		*ltmA* (3.17)
		*marR* (2.39)
		*mtrA* (2.12)

## Data Availability

Data are contained within the article.

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
