# Peer review of "Kanamycin and Ofloxacin Activate the Intrinsic Resistance to Multiple Antibiotics in Mycobacterium smegmatis"

_biology, 2023, doi:10.3390/biology12040506_

Round 1
Reviewer 1 Report
Bacterial infections with antibiotic resistance represent a real threat to human health. Mycobacterium tuberculosis (Mtb) is an example of a deadly pathogen possessing a high ability to tolerate the cytotoxic effects of various drugs via numerous molecular mechanisms leading to the drug resistance (DR). To reduce the rate of resistance evolution, it is vital to understand the complex processes of resistance emergence.
The main goal of the paper by Vatlin et al. was to study the mechanism of intrinsic cross-resistance induced by subinhibitory concentrations of various antibiotics by using M. smegmatis (Msm) as a non-pathogenic proxy of Mtb. Msm is known to possess a wide set of tools to withstand the treatment with various drugs. The authors attempted to investigate the genes for multidrug regulators (belonging to the known Msm resistome) presumably involved in intrinsic cross-DR mediated by non-inhibiting concentrations of antibiotics kanamycin, ofloxacin, streptomycin, tetracycline. The optimal concentration for each antibiotic, which does not affect bacterial growth but can induce cross-resistance, was thoroughly determined. RT-qPCR analyses of changes in expression of the selected genes from the Msm resistome, caused by exposure to antibiotics, allowed the authors to show that the treatment of Msm with kanamycin and ofloxacin at defined non-inhibiting concentrations induced cross-resistance to multiple antibiotics likely due to increase in transcription of the genes whiB7 (for both drugs) and rbpA (for kanamycin). It is an important finding. Although the use of general RNA-seq approach could bring more information, the investigation of the selected genes appeared to be an important step to understanding the mechanisms of multidrug resistance emergence in mycobacteria. The results of this paper endorse the idea that emergence of drug-resistant pathogens may be a consequence of increasing concentrations of various antibiotics in our environment due to uncontrolled use in agriculture, livestock farming and medicine.
Minor points
1. - P3, line 107. Halo diameter was determination… - May be determined?
2. Page 3, paragraph 3.1. The authors wrote that they used following antibiotics as possible inducers - kanamycin, ofloxacin, tetracycline, streptomycin, and erythromycin. At the same time, in Table 1 they presented the data without erythromycin. Explain, please, why erythromycin was excluded from the table and subsequent experiments (Figure 1) and why this drug appears again on page 5 (line 154) as a high-specific inducer (along with streptomycin) that triggers resistance to only rifampicin.
3. Page 7, line 209. Figure 2 demonstrates the relative level of expression for several genes from the Msm resistome in Msm cells cultivated in the presence of microdoses of antibiotics. Expression usually embraces transcription and translation, the authors studied only the transcription level. Thus, it is better to note that ofloxacin behaves as an inhibitor of expression (or transcription) of at least 6 genes (not translation).
.
.
Author Response
Responses to the Reviewer 1:
We would like to say thanks to the Reviewer 1 for the constructive review of our manuscript. The provided comments are very helpful and we have revised the manuscript as suggested. Please find below the point-by-point response to the reviewers’ comments.
“1.- P3, line 107. Halo diameter was determination… - May be determined?.”
Response: Corrected as suggested - “Halo diameter was determined…” (lines 151).
“2. Page 3, paragraph 3.1. The authors wrote that they used following antibiotics as possible inducers - kanamycin, ofloxacin, tetracycline, streptomycin, and erythromycin. At the same time, in Table 1 they presented the data without erythromycin. Explain, please, why erythromycin was excluded from the table and subsequent experiments (Figure 1) and why this drug appears again on page 5 (line 154) as a high-specific inducer (along with streptomycin) that triggers resistance to only rifampicin.”
Response: We removed the data about erythromycin from the manuscript because standard deviations’ values in the measurement experiment «Diameters of growth inhibition halos produced by different antibiotics with and without induction by low-doses of antibiotics on M. smegmatis culture» did not correspond to the criterion chosen by us (intersected). Thank you for pointing out "erythromycin" in the text. We removed “erythromycin” from the text.
“3. Page 7, line 209. Figure 2 demonstrates the relative level of expression for several genes from the Msm resistome in Msm cells cultivated in the presence of microdoses of antibiotics. Expression usually embraces transcription and translation, the authors studied only the transcription level. Thus, it is better to note that ofloxacin behaves as an inhibitor of expression (or transcription) of at least 6 genes (not translation).”
Response: Thanks for the clarification, we've removed "translation" from the text and replaced it with "transcription". (line 522).
We have also made some minor style and typos corrections, and reformatted the references section to include the references added during the revision. We hope the revised manuscript is now acceptable for publication. Thank you for your consideration.
Have a nice day!

Reviewer 2 Report
Because the comprehensive research composition was not logical and the inadequate discussion about all experimental results, this manuscript was so difficult to understand. My comments are listed below.
‧ How did the author decide the MIC value in the analysis of Maximum non-inhibiting concentration?
‧ Why was the Drug susceptibility test not performed according to CLSI or EUCAST recommended methods?
‧ In the gene expression analysis of regulator genes, please show the gene expression levels of M. smegmatis without exposure to antibiotics as a control. Also, two types of genes (sigA and polA) are used as reference genes, but you should indicate how the ΔΔCq values were calculated.
‧ In the drug cross-resistance induction analysis, the authors have made three groups (high, moderate, low) for the obtained results, but should explain how these groups was determined. Furthermore, the author should perform a significant difference test on the diameter results in Fig. 1. Also, the author should show the all diameter results of all drugs and discuss them comprehensively.
‧ The reason for excluding tetracycline exposure in the gene expression analysis is unclear; a specific and detailed explanation of non-specific response is needed. In the first place, it is difficult to understand why the expression analysis did not analyze the expression levels of each antibiotic resistance gene, not just the regulator gene. I believe that clarifying the changes in expression levels of each resistance gene would provide clear evidence of the presence or absence of an effect of Maximum non-inhibiting concentration.
Author Response
Responses to the Reviewer 2:
We would like to say thanks to the Reviewer 2 for the constructive review of our manuscript. The provided comments are very helpful and we have revised the manuscript as suggested. Please find below the point-by-point response to the reviewers’ comments.
“1. How did the author decide the MIC value in the analysis of Maximum non-inhibiting concentration?”
Response: The first step was to determine the MIC of antibiotics in a liquid medium by the method of two-fold dilutions. Then, the second experiment was carried out, in which the MIC value was determined by diluting up and down from the MIC in increments of 0.1 µg/ml for kanamycin and streptomycin and 0.01 µg/ml for ofloxacin and tetracycline. In the next step, we tested this concentration on the agar medium. Section 2. Materials and Methods 2.2 MIC Determination fully describes our experiment and Section 3. Results 3.1. Minimal inhibiting antibiotic concentrations determination for M. smegmatis results of this experiment (line 189). We then reduced the MIC by 2-fold dilutions to 1/32 MIC. A 1/32 MIC concentration of aminoglycosides strongly inhibited the growth of M. smegmatis. For kanamycin and streptomycin, the MIC was diluted 5 times (1/5 MIC), then another 4 (1/20 MIC), 5 (1/25 MIC), 6 (1/30 MIC) times each concentration, after which 2-fold breeding
1/40 MIC, 1/80 MIC in the first case,
1/50 MIC, 1/100 MIC in the second case,
1/60 MIC, 1/120 MIC in the third case,
by measuring the optical density of each sample until the OD600 of each experimental sample is different from the control sample without the addition of antibiotic.
“2. Why was the Drug susceptibility test not performed according to CLSI or EUCAST recommended methods?”
Response: We oriented on CLSI standards contain information about disk diffusion (M02) and dilution (M07, M11) test procedures for bacteria [23] and Laboratory Practices in Microbiology [24]. (line 164).
“3. In the gene expression analysis of regulator genes, please show the gene expression levels of M. smegmatis without exposure to antibiotics as a control. Also, two types of genes (sigA and polA) are used as reference genes, but you should indicate how the ΔΔCq values were calculated.”
Response: The gene expression levels of the experimental samples were normalized to the levels of gene expression in the control samples. Housekeeping genes with constitutive expression sigA and polA were taken as controls. Black line on the figure 2 - M. smegmatis control without induction (Line 264). The ΔΔCq method is detailed in the link [25] (line 176).
“4. In the drug cross-resistance induction analysis, the authors have made three groups (high, moderate, low) for the obtained results, but should explain how these groups was determined. Furthermore, the author should perform a significant difference test on the diameter results in Fig. 1. Also, the author should show the all diameter results of all drugs and discuss them comprehensively.”
Response: The results were divided into three groups according to the number of antibiotics to which resistance was induced: high-specific inducer - to one antibiotic, moderate high-specific inducer - to 2-3 antibiotics, low-specific inducer - to a large number of antibiotics. (line 221). The criterion for selecting positive results in Fig. 1 was the absence of intersections of the standard deviations of the zones of growth inhibition‘s diameters of M. smegmatis samples, which were grown on a medium with an antibiotic (inductor) and on a medium without the addition of an antibiotic. (Line 159).
“5. The reason for excluding tetracycline exposure in the gene expression analysis is unclear; a specific and detailed explanation of non-specific response is needed. In the first place, it is difficult to understand why the expression analysis did not analyze the expression levels of each antibiotic resistance gene, not just the regulator gene. I believe that clarifying the changes in expression levels of each resistance gene would provide clear evidence of the presence or absence of an effect of Maximum non-inhibiting concentration.”
Response: The genomes of mycobacteria contain several dozens of the TetR family (transcriptional repressors) transcriptional regulators. The expression of the genes regulated by them begins after the addition of tetracycline to the medium. Entering into the cell, tetracycline initiates the transcription of many different genes, including those that mediate drug resistance (Lines 233,252). In the future, we plan to study the complete transcriptome of mycobacteria in response to the induction of resistance by microconcentrations of tetracycline.
We have also made some minor style and typos corrections, and reformatted the references section to include the references added during the revision. We hope the revised manuscript is now acceptable for publication. Thank you for your consideration.
Have a nice day!

Reviewer 3 Report
Dear Authors,
I congratulate all the Authors for their contributions to the writing of the manuscript entitled “Kanamycin and ofloxacin activate the intrinsic resistance to multiple antibiotics in Mycobacterium smegmatis”, submitted for publication in the Biology journal.
I have several comments on the manuscript:
1. Line 62: please check the reference (10.1093/infdis/jis191). Formatting error.
2. Line 63: please list the anti-mycobacterial compounds in which Mycobacterium smegmatis is resistant to. What is LOPAC? Please clarify the abbreviation when it is first mentioned in the article.
3. Line 80: what are the concentrations for each antibiotic used in the study?
4. Line 82: there should be a space between a value and its unit (37 °C). Please thoroughly check the whole manuscript for this error.
5. Line 92: a table to define drug susceptibility or resistance to each antibiotic would be an added value to the manuscript (did the Authors follow CLSI guidelines to determine zone of inhibition?).
6. Line 110: g should be written in italics.
7. Table 1: the use of comma and decimal point is inconsistent throughout the table. Please revise this.
8. Figure 1: bacterial names should be in italics.
I consider the manuscript is sufficiently comprehensive for publication in the journal, however it needs a thorough check for its grammar/use of punctuations.
My sincere congratulations to all Authors.
Author Response
Responses to the Reviewer 3:
We would like to say thanks to the Reviewer 3 for the constructive review of our manuscript. The provided comments are very helpful and we have revised the manuscript as suggested. Please find below the point-by-point response to the reviewers’ comments.
“1. Line 62: please check the reference (10.1093/infdis/jis191). Formatting error..”
Response: Link number 17 has been corrected by DOI (Line 399)
“2. Line 63: please list the anti-mycobacterial compounds in which Mycobacterium smegmatis is resistant to. What is LOPAC? Please clarify the abbreviation when it is first mentioned in the article”
Response: Thanks a lot. The abbreviation has been clarified, there is a link in the text [20] (Line 80)/ In this study, it was shown that not all compounds inhibit M. smegmatis and M. tuberculosis at the same time, however, many of the known inhibitors, indicate the presence of similar resistance mechanisms: «For the LOPAC library, 14, 27 and 24 hits were detected against M. smegmatis, M. bovis BCG and M. tuberculosis, respectively. 69, 126 and 91 hits were detected against M. smegmatis, M. bovis BCG and M. tuberculosis, respectively, for the NIH Diversity Set. For the Spectrum Collection, 36, 65 and 45 hits were detected against M. smegmatis, M. bovis BCG and M. tuberculosis, respectively.». It is worth noting that the MIC values for the test substances (ofloxacin, kanamycin, tetracycline, and streptomycin) are similar between the test strain and the TB strain – these data have been added to Table 1 (Line 189).
“3. Line 80: what are the concentrations for each antibiotic used in the study?.”
Response: For the induction we selected the maximum non-inhibiting concentrations of antibiotic, which were founded in this research: kanamycin - 0.03 µg/ml, streptomycin - 0.016 µg/ml, ofloxacin - 0.08 µg/ml, tetracycline - 0.015 µg/ml (Line 199-201). For the disc diffusion tests we selected these concentrations: netilmicin 10 μg/disc, meropenem 10 μg/disc, imipenem 10 μg/disc, spiramycin 100 μg/disc, norfloxacin 10 μg/disc, ofloxacin 5 μg/disc, linezolid 10 μg/disc, kanamycin 30 μg/disc, oxytetracycline 30 μg/disc, azithromycin 15 μg/disc, levofloxacin 5 μg/disc, ciprofloxacin 1 μg/disc, tetracycline 30 μg/disc, lomefloxacin 10 μg/disc, while rifampicin 100 μg/disc was manually impregnated on sterile paper discs. (lines 144-148).
“4. Line 82: there should be a space between a value and its unit (37 °C). Please thoroughly check the whole manuscript for this error.”
Response: Thank you for pointing this out. We have corrected this error (lines103,114, 130, 138, 149).
“5. Line 92: a table to define drug susceptibility or resistance to each antibiotic would be an added value to the manuscript (did the Authors follow CLSI guidelines to determine zone of inhibition?)..”
Response: Thank you for this note. We have added a data table to the appendix of the article. We oriented on CLSI standards contain information about disk diffusion (M02) and dilution (M07, M11) test procedures for bacteria [23] and Laboratory Practices in Microbiology [24] (line 164).
“6. Line 110: g should be written in italics..”
Response: Thank you for pointing this out. We have corrected this error (Line 166).
“7. Table 1: the use of comma and decimal point is inconsistent throughout the table. Please revise this.”
Response: Corrected as suggested. (line 207).
“8. Figure 1: bacterial names should be in italics..”
Response: Corrected as suggested (line 216).
We have also made some minor style and typos corrections, and reformatted the references section to include the references added during the revision. We hope the revised manuscript is now acceptable for publication. Thank you for your consideration.
Have a nice day!

Round 2
Reviewer 2 Report
Thank you for your response to my comments.
